# Greater Durability and Protection against Herpes Simplex Viral Disease following Immunization of Mice with Single-Cycle ΔgD-2 Compared to an Adjuvanted Glycoprotein D Protein Vaccine

**DOI:** 10.3390/vaccines11081362

**Published:** 2023-08-14

**Authors:** Aakash Mahant Mahant, Matthew S. Gromisch, Leah Kravets, Clare Burn Aschner, Betsy C. Herold

**Affiliations:** 1Department of Microbiology-Immunology, Albert Einstein College of Medicine, Bronx, NY 10461, USA; aakash.mahant@einsteinmed.edu (A.M.M.); matthew.gromisch@einsteinmed.edu (M.S.G.); leah.kravets@einsteinmed.edu (L.K.); clare.burnaschner@sickkids.ca (C.B.A.); 2Department of Pediatrics, Albert Einstein College of Medicine, Bronx, NY 10461, USA

**Keywords:** herpes simplex virus, vaccine durability, antibody-dependent cellular cytotoxicity, single-cycle DgD-2 vaccine, glycoprotein D subunit vaccine

## Abstract

Herpes simplex viruses (HSV) cause chronic infections with significant morbidity. Prior vaccines, designed to generate neutralizing antibodies (nAbs) targeting glycoprotein D (gD), failed to provide durable protection. We adopted a different strategy and evaluated a single-cycle virus deleted in gD (ΔgD-2). ΔgD-2elicits antibodies that primarily mediate antibody-dependent cell mediated cytolysis (ADCC) and provides complete protection against clinical isolates of HSV in multiple lethal mouse models. To assess durability, we vaccinated mice (2 doses administered intramuscularly) with ΔgD-2, adjuvanted recombinant gD-2 (rgD-2/Alum-MPL), or uninfected cells as a control, and quantified antibody responses over one year. Mice (n = 5/group) were lethally challenged at 2, 4, 6, 8, and 10-months post-boost. ΔgD-2-vaccinated mice elicited a durable ADCC-mediating response, which provided complete protection against challenge at all timepoints. In contrast, rgD-2/Alum-MPL elicited only nAbs, which declined significantly within 6 months, provided only partial protection at early timepoints, and no protection after 6 months. Serum sampling after viral challenge showed that infection elicited low levels of ADCC-mediating antibodies in rgD-2/Alum-MPL-vaccinated mice and boosted the nAb response, but only after 6 months. Conversely, infection significantly and consistently boosted both the ADCC and nAbs responses in ΔgD-2-vaccinated mice. Results recapitulate clinical trial outcomes with gD vaccines, highlight the importance of ADCC, and predict that ΔgD-2 will elicit durable responses in humans.

## 1. Introduction

Herpes simplex virus types 1 (HSV-1) and 2 (HSV-2) are highly prevalent DNA viruses that infect and replicate in multiple cell types. Both establish latency in sensory neurons with periodic asymptomatic or, less commonly, symptomatic episodes of reactivation that fuel the risk of transmission and disease burden. It is estimated that 3.7 billion people worldwide are infected with HSV-1 and 491 million with HSV-2 [1,2]. HSV-1 is associated with oral and ocular infections and is a leading cause of sporadic infectious encephalitis. HSV-2 is the major cause of primary and recurrent genital and neonatal disease globally, although in the United States and other high-income countries, HSV-1 is emerging as a more common cause of primary genital and neonatal infections [3]. Moreover, HSV is a major driver of the HIV epidemic and is associated with a significantly increased risk for HIV acquisition, transmission, and disease progression [4,5,6]. Mathematical modeling predicts that an HSV-2 vaccine would have a major impact on HIV [5].

Despite its global health impact, efforts to develop an effective HSV vaccine have had limited success. For example, a vaccine comprised of recombinant glycoprotein D (gD-2) formulated with a proprietary aluminum hydroxide (Alum) and monophosphoryl lipid A (MPL) adjuvant, gD2-AS04, administered at 0, 1, and 6 months induced neutralizing antibodies (nAbs) but no antibody-dependent cellular cytotoxicity (ADCC) response [7]. The vaccine failed to protect men or HSV-1 seropositive women against HSV-2 in studies conducted among discordant couples [8]. In a subsequent field trial (Herpevac), where enrollment was restricted to HSV-1 and HSV-2 dually seronegative women, the vaccine provided no protection against genital herpes infection or disease, although in subset analyses modest protection against genital HSV-1 was observed [9,10]. Results of a Phase 1 clinical trial with a replication-defective HSV-2 candidate vaccine strain, dl529, which is deleted in two genes involved in viral replication (*UL5* and *UL29*), were also disappointing [11]. The vaccine elicited ≥4-fold increase in nAbs in HSV seronegative participants but no sustained increase in nAb responses in seropositive participants. Other functional antibody responses were not reported. Moreover, only a subset of participants elicited significant CD4 and even fewer CD8 T cell responses [11].

We have adopted a different strategy and engineered and evaluated a single-cycle HSV-2 virus deleted in the gD-2 gene, the major antigenic target of the previous vaccine trials. In preclinical mouse studies, this candidate vaccine, designated ΔgD-2, elicited high-titer Abs that activate Fc gamma receptors (FcγRs) to induce antibody-dependent cell-mediated cytotoxicity (ADCC), antibody-dependent cellular phagocytosis (ADCP), and complement-dependent cytolysis [12,13,14]. Two doses (prime-boost) administered subcutaneously, intramuscularly, or intradermally completely protected female mice against lethal vaginal challenge and male or female mice from skin challenge [15,16,17,18]. Importantly, we used high doses (10–100× the lethal dose) and clinical isolates of HSV-1 and HSV-2 in these models [15,16,17,18]. In other studies, we showed that vaccination of female mice prior to pregnancy passively protected pups when challenged on Day 7 or 14 of life [19]. Unlike the adjuvanted recombinant gD protein vaccines, ΔgD-2 boosted the total HSV antibody response, elicited new ADCC-mediating Abs in HSV-1 seropositive mice that had recovered from a sublethal infection, and protected the mice from subsequent heterologous challenge with HSV-2 [18]. Protection mapped to the ADCC-mediating Abs, as evidenced by passive transfer studies, which demonstrated that immune serum protected naïve wild-type and complement component 3 (C3) knockout mice but not FcγRIV knockout mice from HSV challenge [14,17]. The importance of FcγRIV-activating Abs was further illustrated by the isolation and characterization of mAbs from the vaccinated mice. We identified an ADCC-mediating mAb that protected mice from vaginal or skin challenges [20].

Another important attribute of vaccines that impacts their efficacy is the durability of the immune response. In the Herpevac trial, the nAbs elicited by gD2-AS04 peaked one month after the 3rd dose at levels similar to those resulting from natural infection but then rapidly declined [21]. This decline may have contributed to the absence of protection [10]. To determine if the mouse model replicated the decline in nAb responses observed in the Herpevac trial and to assess the durability of ADCC-mediating responses to ΔgD-2, we prime-boost vaccinated mice with adjuvanted recombinant gD-2 protein vaccine (rgD-2/Alum-MPL), ΔgD-2, or a control vaccine and quantified antibody responses over one year and vaccine efficacy following lethal skin challenge 2, 4, 6, 8, and 10 months post-boost. 

## 2. Materials and Methods

### 2.1. Cells and Virus

Vero cells, a nonhuman primate epithelial cell line (CCL-81, American Type Culture Collection), and VD60 cells, a Vero cell line transfected with HSV-1 gD that express high levels of the viral protein following viral infection [22], were passaged in Dulbecco’s Modified Eagle Medium (DMEM) (Thermo-Fisher Scientific, Waltham, MA, USA) supplemented with 10% fetal bovine serum (HyClone, Logan, UT, USA) and 1% penicillin-streptomycin (Thermo-Fisher Scientific, Waltham, MA, USA). HSV-2(SD90), a low-passage clinical isolate [23], was propagated on Vero cells, and the viral titer was determined by plaque assay. The vaccine strain ΔgD-2 has been previously described [24] but was modified to remove the green fluorescent protein (provided by William R. Jacobs, Albert Einstein College of Medicine). The vaccine viral strain was grown on VD60 cells, and viral titers were determined by plaque assay on complementing and non-complementing Vero cells. HSV-2 recombinant gD-2 protein (rgD-2) was provided by the Einstein Protein Core Facility and resuspended in 150 µg of alum (Imject-Thermo Scientific, Thermo-Fisher Scientific, Waltham, MA, USA) and 12.5 µg of monophosphoryl lipid A (MPL) (Invivogen, San Diego, CA, USA) [25].

### 2.2. Immunization and Infection of Mice

The study was approved by the Institutional Animal Care and Use Committee at the Albert Einstein College of Medicine, protocol 0000-1425. Female C57BL/6J mice, purchased from Jackson Laboratory (JAX, Bar Harbor, ME, USA), were vaccinated intramuscularly (prime) at 6 weeks of life and then three weeks later boosted with 5 × 10^6^ plaque-forming units (pfu) of ΔgD-2-infected VD60 cell lysate, control VD60 cell lysate, or 5 μg of adjuvanted gD-2 protein (rgD-2/Alum-MPL). The dose and route of immunization were chosen based on prior dose escalation studies [16]. We used uninfected VD60 cells as the comparator since gD is only expressed after HSV infection, and thus the cells provide a control for cellular proteins. Vaccinated mice were infected on the skin (after depilation and scarification as previously described [13]) with 10× the LD90 of HSV-2(SD90) (~1 × 10^6^ pfu/mouse) 2, 4, 6, 8, and 10-months post-boost, which corresponds to mouse chronological ages of ~4, 6, 8, 10, and 12 months. Mice were monitored and scored daily for disease as follows: (0) no signs of infection; (1) erythema at the inoculation site; (2) localized lesions, edema, and/or spread of erythema; (3) further distal spread, ulcerations, and/or hind limb weakness or paresis; (4) hind limb paralysis; and (5) death. Mice were euthanized at a score of 4 and assigned a score of 5 the following day. Surviving mice were euthanized on Day 14. Blood was collected by retro-orbital bleeding one week post-boost, one week prior to each infection, 14 days post-infection in survivors, or at the time of euthanasia in mice with disease scores ≥ 4. Serum was separated, divided into aliquots, and stored at −80 °C for immunological assays. Sciatic nerve tissue was collected at the time of euthanasia and stored at −80 °C for DNA isolation.

### 2.3. Antibody Assays

Total HSV-infected cell lysate binding and gD-binding IgG were measured in serum by enzyme-linked immunosorbent assays (ELISA) as previously described [16,25]. Briefly, plates were coated with lysates of Vero cells that had been infected for 24 h with HSV-2(SD90) at a multiplicity of infection (MOI) of 0.1 pfu/cell, uninfected Vero cells, or gD protein (10 ng/well). Serial dilutions of vaccinated mouse serum were then added to duplicate wells and incubated overnight at 4 °C, and bound IgG was quantified using horseradish peroxidase (HRP)-conjugated secondary antibodies (Thermo Scientific, Waltham, MA, USA). The results are presented as the optical density value measured using a microplate reader with a 450 nm filter (OD450) at the indicated serum dilutions. 

Neutralizing titers were calculated using a plaque-reduction assay. Virus (100 pfu of SD90) was incubated with serial 2-fold dilutions of heat-inactivated serum (56 °C for 30 min) pooled from each group of 5 mice for 1 h prior to inoculating duplicate wells of Vero cells that had been grown on a 24-well plate. After 1 h of incubation at 37 °C, the inoculum was removed, and the cells were overlaid with medium containing 0.5% methyl cellulose for 48 h at 37 °C. Cells were fixed with methanol for 15 min and stained with Crystal Violet. Plaques were counted, and the neutralization titer was defined as the highest dilution, resulting in a 50% reduction in plaques compared to control wells (virus incubated with media only). 

FcγRIV activation, a biomarker of ADCC, was measured using the mFcγRIV ADCC Reporter Bioassay (Promega, Madison, WI, USA) as previously described [12]. Vero cells were infected with HSV-2(SD90) overnight at an MOI of 0.1. These target cells were transferred to white, flat-bottomed 96-well plates, and duplicate wells were incubated with a 1:5 dilution of serum pooled from each group of 5 mice for 15 min. Reporter cells were added at a target: effector cell ratio of 1:25 for 6 h at 37 °C, 5% CO_2_. FcγRIV activation was detected by luciferase luminescence using a SpectraMax M5^e^ (Molecular Devices). Fold induction was calculated relative to luciferase activity in the absence of serum.

### 2.4. Detection of HSV Viral DNA in Neuronal Tissue by Quantitative Polymerase Chain Reaction

DNA was isolated from sciatic nerve tissue using the DNeasy Blood and Tissue Kit (Qiagen, Hilden, Germany) and the DNA concentration was determined by Nanodrop. We used 10 ng of total DNA per reaction. To quantify HSV DNA, we performed real-time quantitative PCR (RT-qPCR) with the TaqMan Gene Expression Master Mix (Thermo Scientific, Waltham, MA, USA). Primers and probes for HSV-2 DNA polymerase (*U_L_30*) were purchased from Integrated DNA Technologies (Coralville, IA, USA) [13]. Quantified HSV-2 viral DNA was used as a standard curve. Samples with fewer than four copies detected were considered negative. β-Actin was amplified as a control.

### 2.5. Statistical Analysis

We used GraphPad PRISM version 10.0.0 (GraphPad Software, San Diego, CA, USA) for statistical analyses and the generation of graphs. We considered a *p*-value of 0.05 to be statistically significant. Disease scores were compared using a two-way ANOVA with Sidak’s multiple comparisons test. Other results were compared with one-way ANOVA with indicated multiple comparison tests, paired or unpaired two-tailed *t* tests, as indicated.

## 3. Results

### 3.1. Persistence of HSV Antibody Responses to ΔgD-2 but not Adjuvanted gD Protein Vaccination

Mice were prime-boost vaccinated with ΔgD-2, rgD-2/Alum-MPL, or, as a control, an uninfected VD60 cell lysate. Blood was obtained and serum assayed for antibody responses one week post-boost (n = 25 mice per vaccination group) and then one week prior to each viral challenge with HSV-2(SD90) on the skin (e.g., ~2, 4-, 6-, 8-, and 10-months post-boost; n = 5 mice/group/time point), as illustrated in Figure 1. Blood and sciatic tissue were also collected 14 days after the viral challenge or earlier if the mice succumbed to the infection. 

Both rgD-2/Alum-MPL (Figure 2A) and ΔgD-2 (Figure 2B) elicited an HSV-2 lysate antibody response, but the response to rgD-2/Alum-MPL was significantly less than the response to ΔgD-2 (*p* < 0.0001, unpaired *t*-test), began to decline within 6 months, and was significantly reduced compared to the immediate post-boost levels at 8 and 10 months (*p* < 0.05, paired *t*-test). In contrast, there was no significant decline in HSV-2 lysate-binding antibodies elicited by ΔgD-2 at any time point. Conversely, the rgD-2/Alum-MPL vaccine elicited a significantly greater gD-specific antibody response (Figure 2C) compared to ΔgD-2 (Figure 2D) (*p* < 0.0001), which elicited little anti-gD-specific antibody, reflecting the fact that gD is not expressed by progeny viruses. However, corresponding with the decrease in total HSV-2 antibodies, the gD-specific antibody levels declined significantly in the rgD-2/Alum-MPL-vaccinated mice as early as 6 months post-boost (*p* < 0.01 at 6 and 8 months and *p* < 0.0001 at 10 months). The OD signal for the antibodies generated in response to rgD-2/Alum-MPL showed little decline between the 1:1000 and 1:10,000 dilutions, which may reflect a matrix effect specific to gD antibodies. 

### 3.2. Differences in Functionality and Durability of Responses to ΔgD-2 Compared to rgD-2/Alum-MPL Vaccination

Consistent with prior clinical studies, rgD-2/Alum-MPL elicited a neutralizing response (mean log_2_ neutralization titer 6.83 ± 0.06 post-boost), which was significantly greater than the post-boost response in ΔgD-2 vaccinated (4.85 ± 0.06) or control-vaccinated (3.42 ± 0.16) mice (*p* < 0.0001, ANOVA) (Figure 3A). The low level of neutralizing activity detected in serum from VD60 lysate control-vaccinated mice could reflect antibodies elicited by Vero cells or other factors in the serum, as there were no anti-HSV or anti-gD antibodies detected by ELISA (Appendix A). Conversely, ΔgD-2 elicited significantly greater ADCC-mediating responses (26.5 ± 0.9-fold increase in FcγRIV activation) compared to rgD-2/Alum-MPL (0.6 ± 0.1) or control (0.4 ± 0.1) vaccinated mice (*p* < 0.0001) (Figure 3B). There was a decline in neutralizing titers in the rgD-2/Alum-MPL-vaccinated mice, which were significantly lower than the post-boost titer at 6- (*p* < 0.01), 8- (*p* < 0.05), and 10- (*p* < 0.05) months post-boost. In contrast, there was no significant decline in ADCC-mediating elicited by ΔgD-2. Moreover, although the nAb response to ΔgD-2 was significantly lower than that elicited by rgD-2/Alum-MPL, it also persisted over time. 

### 3.3. ΔgD-2 Provides Significantly Greater and More Durable Protection Compared to rgD-2/Alum-MPL against HSV-2 Disease in Mice

To determine if the differences in antibody function and durability translated into differences in disease protection, groups of vaccinated mice (n = 5 per group) were challenged on the skin with 10x the LD90 of HSV2(SD90). At each time point, 100% of the ΔgD-2 vaccinated mice survived, with a peak disease score of 1, except at 4 months, where 3 mice had a peak disease score of 2, which quickly resolved (*p* < 0.0001 compared to control and rgD-1/Alum-MPL vaccinated mice at all timepoints). In contrast, the rgD-2/Alum-MPL vaccine provided only partial protection when the mice were challenged 2 months and 4 months post-boost (*p* < 0.01 compared to VD60-vaccinated controls) but provided no survival benefit at later timepoints (Figure 4). 

No virus was detected in any of the sciatic nerve tissue harvested on Day 14 post-challenge (time of euthanasia) in the ΔgD-2 vaccinated mice until 10 months post-boost, when 4/5 mice had low (<10^3^ copies/10 ng DNA) but significantly less HSV DNA detected compared to rgD-2/Alum-MMPL or control vaccinated mice (*p* < 0.01 and *p* < 0.05, respectively) (Figure 5). In contrast, rgD-2 vaccination only resulted in a significant decrease in the quantity of HSV DNA isolated from sciatic nerve tissue compared to control mice at 2 months and 6 months post-boost (*p* < 0.001), but not at other timepoints. 

### 3.4. Infection Boosts Preexisting Antibodies and Generates New Functional Responses

To evaluate whether infection boosted vaccine responses or elicited new antibody responses, serum was obtained on Day 14 in survivors or at the time of demise in mice with disease progression and compared to results obtained from serum assayed one week prior to each viral challenge. In control (VD60) vaccinated mice, infection elicited a statistically significant increase in HSV-2-lysate binding, gD binding, and nAb responses within one week. Infection also elicited low levels of ADCC-mediating Abs that were significantly greater than pre-infection levels at 4 months and 10 months (Figure 6A). In the rgD-2/Alum-MPL-vaccinated mice, infection boosted the total HSV-2 lysate binding ELISA antibody responses at all time points but only boosted the gD binding and nAb responses after 6 months (Figure 6B), which coincides with the decline in vaccine-induced responses (Figure 2 and Figure 3). Infection also elicited an increase in ADCC-mediating antibodies at each timepoint, although the magnitude of the response resembled (and was not significantly different from) the post-infection increase observed in control vaccinated mice and was significantly less than the response elicited in the ΔgD-2 vaccinated mice (Figure 6C). Viral challenge in ΔgD-2 vaccinated mice boosted HSV binding, gD-specific, neutralizing, and ADCC-mediating Abs responses at all timepoints, although the effects on total HSV-lysate binding Abs were more modest and not significant at every timepoint. Notably, the ADCC-mediating antibodies increased from a mean ± SEM of 26.5 ± 0.9 to 39.9 ± 0.7 following the viral challenge.

## 4. Discussion

Most vaccines for the prevention of HSV continue to focus on neutralizing anti-gD antibodies as a correlate of protection, but this approach has been challenged by disappointing clinical trial results. The biological basis for these vaccine failures is not well defined but may reflect the need for a more potent and durable neutralizing response and/or the need to redefine correlates of protection and develop vaccines that elicit antibodies capable of mediating ADCC, other non-neutralizing functions, and/or a greater cytolytic T cell response [26]. The Herpevac trial of adjuvanted gD protein subunit vaccine elicited neutralizing antibodies but not ADCC and failed to prevent HSV-2 infection or disease [7,10]. Moreover, the neutralizing titer declined rapidly from the peak of log_2_ 7.6 one month after the 3rd dose to log_2_ 5.1 five months later [21].

Several aspects of the current mouse studies recapitulate the Herpevac clinical trial results, although the mice received only two doses of adjuvanted gD-2 (5 μg each combined with Alum and MPL three weeks apart) compared to study participants, who received three doses (20 μg adjuvanted with AS04 at months 0, 1, and 6). Despite the different dosing schedules, the gD protein vaccine elicited comparable neutralizing responses in humans and mice that peaked at similar levels (log_2_ 7.6 and log_2_ 6.8, respectively), and declined within 6 months of the final dose. The gD protein vaccine failed to elicit ADCC-mediating responses in mice or humans [7]. Moreover, while the subunit protein vaccine initially provided incomplete protection against a lethal challenge with a clinical isolate of HSV-2 in the mice, no protection was observed after 6 months, findings that reflect the clinical trial results with HSV-2 [10]. The precise inoculum needed to infect humans is not known and is likely to be less than the dose used in this lethal model. However, our findings suggest that this stringent model in which the mice are challenged with a high dose of a clinical isolate of HSV-2 (SD90) on the skin may prove more predictive of clinical trial outcomes compared to earlier mouse or guinea pig models where animals were typically challenged intravaginally with laboratory-adapted viral strains (e.g., HSV-2(MS) [27,28,29].

The results with the ΔgD-2 vaccine differed markedly from those with rgD-2/Alum-MPL in this model. First, consistent with prior studies, this single-cycle candidate viral vaccine elicited a predominantly ADCC-mediating response but only low levels of nAbs [12,13]. Second, the antibody responses persisted, with no significant decline at 10 months post-boost (chronological age ~12 months). The response completely protected mice from high-dose challenges throughout the study period, and no virus was detected in the sciatic nerve until 10 months post-boost. The ability to detect low levels of HSV DNA in neuronal tissue at this last timepoint could reflect the small but not statistically significant decline in ADCC-mediating antibodies and/or an age-associated decrease in effector cell killing function that allowed some viruses to spread to neuronal tissue. Decreased NK cell activity has been described in elderly adults and has been suggested to contribute to an increase in the incidence and severity of viral infections [30]. 

We elected to evaluate a relatively high dose of ΔgD-2 in this study based on our prior dose escalation studies [16]. This dose is comparable to doses used in mice [23] and Phase 1 clinical studies of the replication-defective virus, dl5-29. In the Phase 1 clinical trial, participants were immunized with 10^7^ pfu of virus/dose at 0, 1, and 6 months [11]. We recognize that dose and route of vaccination may impact durability, and this will need to be evaluated in clinical studies. 

The importance of vaccine durability has been highlighted by the recent experience with COVID-19, where protection from natural infections and vaccines has waned. Precisely what determines the durability of protective antibody responses is unclear and likely reflects pathogen factors (e.g., inoculum size, frequency of exposure, and incubation time) as well as the host immune response, including the quantity and quality of antibodies, epitope specificity, breadth, and isotype [31]. Durability is associated with a prolonged germinal center B cell response and increased B cell somatic hypermutation [31]. Both the ADCC-mediating antibodies and the low level of nAbs generated by ΔgD-2 persist much longer than the anti-gD neutralizing responses elicited by rgD-2, suggesting a more sustained germinal center response to antigens presented by ΔgD-2. This difference may be mediated in part by the signaling molecular switch molecule, herpes virus entry mediator (HVEM). Glycoprotein D blocks interactions between HVEM and its natural ligands, and we have previously published that HVEM signaling is needed for the generation of protective ADCC-mediating antibodies [17]. We speculate that intact HVEM signaling also allows for the generation of a more durable immune response, a notion that is supported by studies showing that HVEM modulates the stringency of germinal center B cell selection [32]. 

The current study also provided the opportunity to assess how viral exposure modulates vaccine responses. Viral challenge boosted the gD-specific and nAbs titers in the rgD-2/Alum-MPL vaccinated, but only after vaccine titers began to fall. Infection also elicited a low level of ADCC-mediating antibodies similar in quantity to those elicited in the control-vaccinated mice. In contrast, exposure of the ΔgD-2 vaccinated mice to the virus increased both the nAb and ADCC responses at all time points. This rapid boosting likely reflects a reservoir of memory B cells that contribute to the durability of the protective responses. The increased antibody response following viral challenge suggests that ΔgD-2 may be effective as a therapeutic vaccine and limit the spread of reactivating viruses.

In conclusion, the results of these studies support the use of this mouse model, which uses high-dose challenges and clinical isolates of HSV-2, in the preclinical evaluation of candidate vaccines. The findings also suggest that ADCC-mediating antibodies may prove to be a more predictive correl ate of immune protection and support the advancement of ΔgD-2 as a novel candidate vaccine that may elicit durable immune responses.

## Figures and Tables

**Figure 1 vaccines-11-01362-f001:**
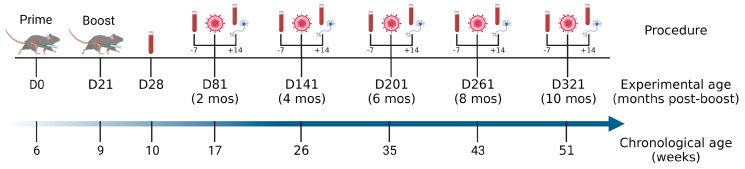
Experimental timeline and chronological age of mice. The mice were primed and boosted with vaccines at 6 and 9 weeks of age, respectively, and then 2, 4, 6, 8, and 10 months later were challenged with 10× the lethal dose of HSV-2(SD90) on the skin. The time of additional blood collection (7 days before infection; -7) and blood and sciatic nerve tissue collection (14 days after infection or at the time mice succumbed to disease; +14) is indicated.

**Figure 2 vaccines-11-01362-f002:**
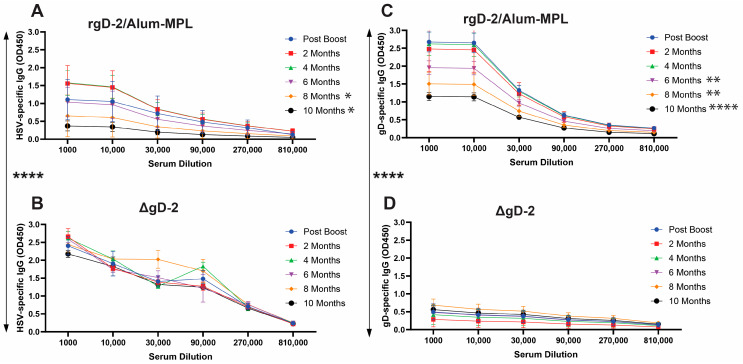
Antibody titers persist in ΔgD-2 vaccinated mice but decline after 6 months in rgD-2/Alum-MPL-vaccinated mice. Serial dilutions of serum obtained one week and then 2, 4, 6, 8, and 10-months post-boost from rgD-2/Alum-MPL (**A**,**C**) or ΔgD-2 (**B**,**D**) vaccinated mice were analyzed for HSV-lysate binding (**A**,**B**) or gD-protein binding (**C**,**D**) antibodies by ELISA. The OD450 at each dilution (mean ± sd, n = 5 mice per group) is shown, and the area under the dilution curve (AUC) is compared at each time point to the post-boost AUC (paired) and between the different vaccines post-boost (unpaired) t-tests (* *p* < 0.05, ** *p* < 0.01, **** *p* < 0.0001).

**Figure 3 vaccines-11-01362-f003:**
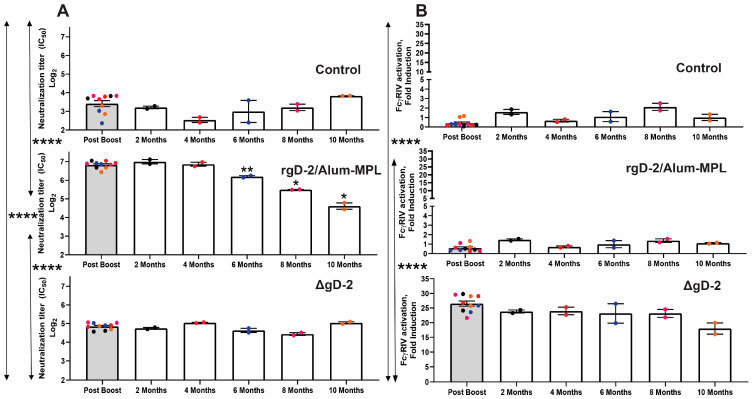
ADCC-mediating and neutralizing antibodies elicited by ΔgD-2 persist, whereas rgD-2/Alum-MPL elicits only neutralizing responses that decline after 6 months. Serum was pooled from n = 5 mice/vaccine group at each time point and analyzed in duplicate for neutralizing activity and mFcγRIV activation. (**A**) The neutralizing titer is presented as a log2 titer that inhibited 50% of viral plaques compared to no serum (mean ± SEM). (**B**) The mFcγRIV activation was assayed at a 1:5 dilution of serum and is presented as fold change relative to no serum (mean ± SEM). For each vaccination group, the responses were compared relative to post-boost at each time point by paired t-test, and the response post-boost was compared between the different vaccine groups by ANOVA (* *p* < 0.05, ** *p* < 0.01, **** *p* < 0.0001). Each group of mice is color coded.

**Figure 4 vaccines-11-01362-f004:**
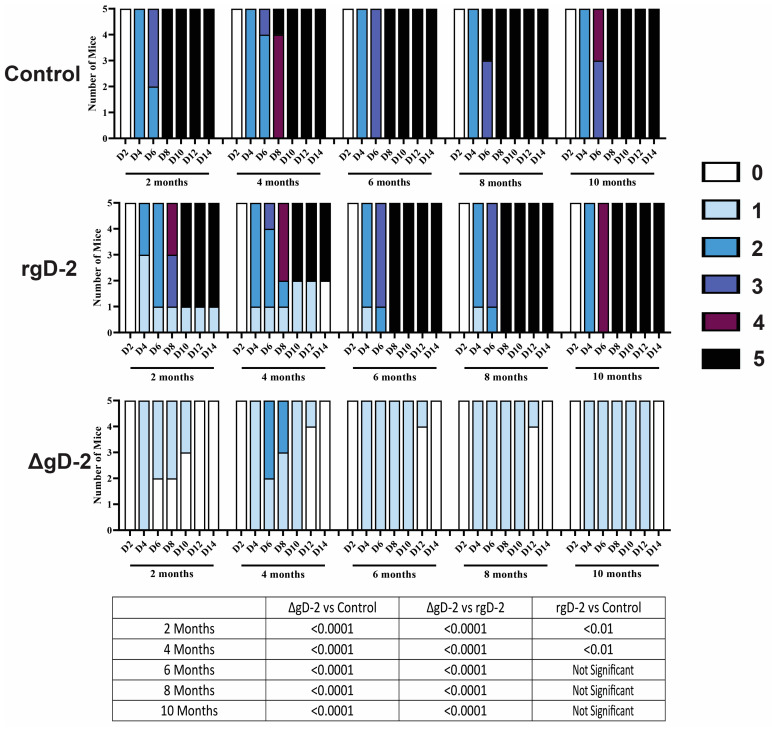
ΔgD-2 protects against lethal challenge up to 10 months post-boost. Vaccinated mice (n = 5 per group) were challenged on the skin with 10× LD90 of SD90 at the indicated times post-boost and scored for signs of disease as indicated in Methods. Disease scores were compared by ANOVA at each timepoint, as indicated in the Table.

**Figure 5 vaccines-11-01362-f005:**
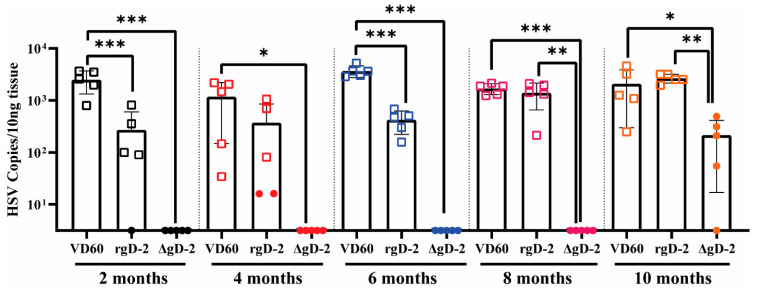
ΔgD-2 prevents the virus from reaching neuronal tissue. The sciatic nerve innervating the area of skin infection was harvested on Day 14 post-challenge in mice who survived (closed symbols) or at the time of euthanasia in those who succumbed to disease (open symbols) and assayed for the presence of HSV DNA by quantitative PCR. Each symbol represents the HSV DNA copies/10 mg of tissue for individual mice, and the line indicates the mean ± SD. At each time point, groups were compared by one-way ANOVA with Dunn’s multiple comparison test (* *p* < 0.05, ** *p* < 0.01, *** *p* < 0.001).

**Figure 6 vaccines-11-01362-f006:**
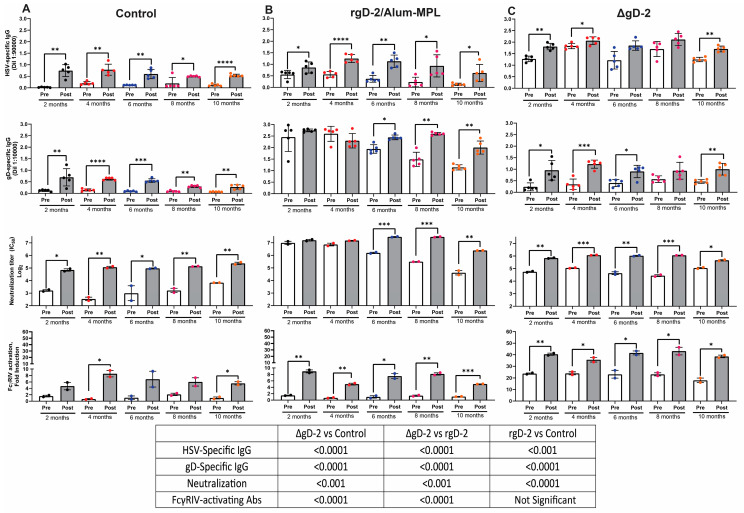
Infection boosts or elicits new antibody responses in vaccinated mice. HSV lysate binding, gD-binding, neutralizing, and FcγRIV-activating antibodies were quantified in post-infection serum (obtained at the time of euthanasia) and compared to responses one week prior to infection at each time point in control (VD60) (**A**), rgD-2/Alum-MPL (**B**), and ΔgD-2 vaccinated mice (**C**). The HSV-binding and gD-binding titers are presented at the indicated dilution (mean ± SD), nAbs as log2 titers that inhibited 50% of viral plaques compared to no serum, and FcγRIV-activating Abs as fold-induction at a serum dilution of 1:5 (both mean ± SEM, pooled serum, n = 5 mice/group). The pre- and post-infection samples are compared by paired *t*-test, and responses between vaccine groups are compared by ANOVA (* *p* < 0.05, ** *p* < 0.01, ****p* < 0.001, and **** *p* < 0.0001) (Table).

## Data Availability

All data supporting the findings of this study are available upon request.

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
