# Peer review of "Greater Durability and Protection against Herpes Simplex Viral Disease following Immunization of Mice with Single-Cycle ΔgD-2 Compared to an Adjuvanted Glycoprotein D Protein Vaccine"

_vaccines, 2023, doi:10.3390/vaccines11081362_

Round 1
Reviewer 1 Report
In this study, Mahant et al. demonstrate the superior durability and efficacy of a single-cycle ΔgD-2 vaccine compared to an adjuvanted glycoprotein D subunit vaccine (rgD-2/Alum-MPL) against lethal herpes simplex virus 2 (HSV-2) challenge in a mouse model. The ΔgD-2 vaccine induced robust antibody-dependent cellular cytotoxicity (ADCC) that persisted and fully protected mice from lethally challenge at 10 months post-vaccination. In contrast, rgD-2/Alum-MPL elicited neutralizing antibodies that waned after 6 months and conferred only transient protection, mirroring previous failed clinical trials. These findings provide evidence that durable ADCC-mediating antibodies are imperative for HSV immunity and should be a priority for next-generation vaccine design. Overall, this is a well-designed study that offers key insights into the limitations of previous gD vaccines and highlights a path forward using vaccines that generate lasting ADCC response against HSV.
I only have a few minor comments as follows.
- Why did the authors use VD60 cell lysate as the control for mouse vaccination?
- Figure 1: The scheme is somewhat confusing regarding the blood sampling time points around the lethal challenge. Please simplify it for better clarity and ease of comprehension. In addition, important details about the viral challenge experiments, the specific name of the HSV-2 isolate used (e.g. HSV-2(SD90)) and the route of administration (e.g. skin infection) should be clearly stated in the text (line 166) or in the figure.
- Figure 2: It is noteworthy that the OD450 signals show little decrease after a 10-fold dilution of serum from 1000 to 10000, which appears unusual.
- Figure 3B: The Y-axis scales for the upper two graphs should be adjusted to enable readers to observe the differences between groups more clearly.
- The abbreviation "odu" should be defined as "optical density units" when first used in line 190.
Author Response
Response to Reviewer 1:
We thank the Reviewer for the suggestions and minor comments. We have made the suggested changes as described below:
- Why did the authors use VD60 cell lysate as the control for mouse vaccination?
- We used VD60 lysate as the control since the delta gD vaccine is grown on this cell line and thus it provides a control for cellular antigens. We added a sentence to the Methods clarifying this.
- Figure 1: The scheme is somewhat confusing regarding the blood sampling time points around the lethal challenge. Please simplify it for better clarity and ease of comprehension. In addition, important details about the viral challenge experiments, the specific name of the HSV-2 isolate used (e.g. HSV-2(SD90)) and the route of administration (e.g. skin infection) should be clearly stated in the text (line 166) or in the figure.
- We modified Figure 1 schema for greater clarity and also added the route and clinical isolate used for viral challenges as suggested.
- Figure 2: It is noteworthy that the OD450 signals show little decrease after a 10-fold dilution of serum from 1000 to 10000, which appears unusual.
- As the reviewer noted, the odu signal for the antibody response to rgD-2/Alum-MPL showed little decline between the first two dilutions (1:1000 and 1:10,000). This likely reflects a matrix effect specific to anti-gD antibodies, which is overcome by further dilutions. In contrast, the odu signal for antibodies elicited by the gD-2 deletion virus declined with each subsequent dilution. We added a comment to this effect in the results section.
- Figure 3B: The Y-axis scales for the upper two graphs should be adjusted to enable readers to observe the differences between groups more clearly.
- We adjusted the y-axis into 2 segments to enable readers to observe differences more clearly.
- The abbreviation "odu" should be defined as "optical density units" when first used in line 190.
- We changed this to OD and defined it in Methods section.
Reviewer 2 Report
The manuscript describes a novel HSV vaccine based on HSV deleted in gD. Results clearly show a superior immunogenicity and effectiveness, in a murine model of HSV-2 challenge, compared to the recombinant gD protein based vaccine. The authors rightfully discuss these results in the context of development of a novel HSV vaccine to address the failure of previous attempts.
Major comment:
The main limitation of the study is linked to the evaluation of a single dosage of delta gD HSV vaccine. The dosage of vaccine selected for the studies of 5 X 10e6 pfu is a quite high dosage for mice. It is crucial to understand if this approach has a translational value by evaluating effectiveness for one or two lower dosages like for example 1 X 10e6 pfu and 5 X 10e5 pfu.
Minor comments:
Data representation should be improved.
i) figure legends should be more concise
ii) data should be represented in a canonical way.
figure 2
should have on the X axis the time course instead of serum dilutions: moreover it would be preferable to show the antibody tire calculated on the dilution instead of the OD values at all dilutions.
figure 3
curves are a better representation of changes over time than individual columns
figure 6
showing changes in the calculated antibody titres would be more appropriate than showing OD values at one selected dilution
Round 2
Reviewer 2 Report
new version is fine